# Embryonic and foetal expression patterns of the ciliopathy gene *CEP164*

L. A. Devlin[1], S. A. Ramsbottom[1], L. M. Overman[2], S. N. Lisgo[2], G. Clowry[3], E. Molinari[1], L. Powell[1], C. G. Miles[1], J. A. Sayer[1,4,5]*

**1** Institute of Genetic Medicine, Newcastle University, Central Parkway, Newcastle upon Tyne, England, United Kingdom, **2** MRC-Wellcome Trust Human Developmental Biology Resource, Institute of Genetic Medicine, International Centre for Life, Newcastle upon Tyne, England, United Kingdom, **3** Institute of Neuroscience, The Medical School, Newcastle University, Framlington Place, Newcastle upon Tyne, England, United Kingdom, **4** The Newcastle upon Tyne Hospitals NHS Foundation Trust, Freeman Road, Newcastle upon Tyne, England, United Kingdom, **5** National Institute for Health Research Newcastle Biomedical Research Centre, Newcastle upon Tyne, England, United Kingdom

* john.sayer@ncl.ac.uk

**Data Availability Statement:** All relevant data are within the paper.

**Funding:** L.A.D is funded by the Medical Research Council Discovery Medicine North Doctoral

## Abstract

Nephronophthisis-related ciliopathies (NPHP-RC) are a group of inherited genetic disorders that share a defect in the formation, maintenance or functioning of the primary cilium complex, causing progressive cystic kidney disease and other clinical manifestations. Mutations in centrosomal protein 164 kDa (*CEP164*), also known as *NPHP15*, have been identified as a cause of NPHP-RC. Here we have utilised the MRC-Wellcome Trust Human Developmental Biology Resource (HDBR) to perform immunohistochemistry studies on human embryonic and foetal tissues to determine the expression patterns of CEP164 during development. Notably expression is widespread, yet defined, in multiple organs including the kidney, retina and cerebellum. Murine studies demonstrated an almost identical *Cep164* expression pattern. Taken together, these data support a conserved role for CEP164 throughout the development of numerous organs, which, we suggest, accounts for the multi-system disease phenotype of *CEP164*-mediated NPHP-RC.

## Introduction

Nephronophthisis-related ciliopathies (NPHP-RC) are a collection of inherited genetic disorders, grouped together by a core defect in the formation, maintenance or functioning of the primary cilium complex [1, 2]. NPHP-RC patients typically present with nephronophthisis, a fibrotic cortico-medullary cystic kidney phenotype, which frequently leads to end stage-renal disease (ESRD) [3, 4]. In some NPHP-RC cases, including Senior-Løken syndrome (SLSN), Alström syndrome (AS), Bardet Biedl syndrome (BBS) and Joubert syndrome (JBTS), patients have retinal dysplasia and degeneration phenotypes, such as Leber congenital amaurosis, which can deteriorate to blindness [5–7]. Neurological abnormalities are often present; JBTS patients have midbrain cerebellar vermis hypoplasia, characterised by the "molar tooth" sign on MRI analysis [5, 8], leading to numerous problems including ataxia, hypotonia and breathing abnormalities. Intellectual disability and developmental delay, are also demonstrated

Training Partnership (https://www.dimen.org.uk) and The Northern Counties Kidney Research Fund (http://www.nckrf.org.uk), S.A.R is supported by a Kidney Research UK Post-doctoral fellowship (PDF_003_20151124). JAS is funded by Kidney Research UK and The Northern Counties Kidney Research Fund. The funders had no role in study design, data collection and analysis, decision to publish, or preparation of the manuscript.

**Competing interests:** The authors have declared that no competing interests exist.

throughout the spectrum of NPHP-RC. Additionally, BBS patients are often diagnosed with hypogonadism and/or obesity [9]. Severe NPHP-RC phenotypes, including Jeune syndrome and Meckel Gruber syndrome display skeletal dysplasia, and lethal occipital encephalocele. Consistent with ciliopathy syndromes, polydactyly, liver fibrosis, facial dysmorphism, cardiac abnormalities, hearing loss and type 2 diabetes can present as secondary symptoms [9–11]. Although some disease management therapies are available, notably, there is no cure for NPHP-RC.

There are at least 25 NPHP-RC causative genes currently identified [12], accounting for a molecular genetic diagnosis in around 60% of NPHP-RC patients. Mutations in *CEP164*, also known as *NPHP15*, have been identified as a cause of NPHP-RC [1, 9]. These patients have a largely heterogenous clinical presentation. The majority suffer with NPHP and a retinal phenotype, which in some patients causes blindness prior to two years of age. One patient was diagnosed with JBTS, and two others have BBS-like manifestations with an absence of renal problems [1, 9]. It has been proposed that homozygous truncating mutations of *CEP164* cause a severe JBTS phenotype, whereas hypomorphic mutations may cause a milder syndromic presentation [1]. Patients with identical *CEP164* mutations can display different clinical manifestations, which makes it difficult to understand the pathology and progression of the disease.

*CEP164* was first identified and cloned from an adult foetal brain cDNA library [13] and is located on chromosome 11, q23.3. Its largest, most commonly cited isoform is 5,629 bp, encoding the 1,460 amino acid protein (CEP164) (NM_014956) [1, 14]. There is an alternative isoform of CEP164 (NM_001271933), with a 1,455 amino acid product [1, 13, 14]. CEP164 is a centrosomal protein that localises to each of the nine distal appendages of the primary cilia mature centriole, in a microtubule-independent manner [1, 2, 15–22]. This has been demonstrated by super-resolution microscopy techniques in multiple human and murine cell lines. Cell-cycle dependent recruitment of CEP164 to the distal appendages is hierarchical, within a network of other distal appendage proteins. These include CEP83 (mutations in *CEP83* cause NPHP18), CEP89, SCLT1 (variants in SCLT1 may be associated with Orofaciodigital syndrome 9), and FBF1. CEP164 is recruited last, defining the proximal end of the transition zone [2, 15, 18, 22–26].

Protein domain analysis has identified that CEP164 has an N-terminal tryptophan-tryptophan (WW) domain, an area of lysine-rich repeats (LR), and multiple serine-glutamine/threonine-glutamine (SQ/TQ) potential phosphorylation sites [2, 14]. There are at least 3 predicted coiled-coil domains [1, 2, 14, 15] and the C-terminal domain is currently undefined (**S1 Fig**). Multiple fusion protein studies have demonstrated that the CEP164 C-terminal domain is required for localisation of CEP164 to the distal appendages [1, 15]. In some studies, CEP164 has been found within the nucleus [1, 14, 17, 27].

Numerous in vitro *CEP164* knockdown studies (siRNA/CRISPR) have demonstrated that *CEP164* loss causes aberration of ciliogenesis, with disruption of primary cilia production prior to transition zone formation; notably centriolar structure is not disrupted [15, 23, 25, 28, 29]. Upon initiation of ciliogenesis Rabin 11 imports the GTPase Rabin 8 to the basal body. CEP164 interacts with Rabin 8, facilitated by Chibby 1, to recruit Rab8 positive vesicles to the centrosome [15, 17, 19, 24, 30]. Vesicle docking is required for subsequent basal body anchoring to the plasma membrane and primary cilia development. Additionally, CEP164 forms a complex with tau tubulin kinase 2 (TTBK2) via its N-terminal WW domain. CEP164 recruits TTBK2 to the primary cilia basal body, which allows removal of the centriolar capping protein, CP110, potentially via phosphorylation, and then initiation of intraflagellar transport recruitment with subsequent axonemal extension [1, 15, 29, 31, 32]. The CEP164/Rabin 8 and CEP164/TTBK2 pathways work independently of each other [29]. There are other predicted interactors of CEP164 including dishevelled, NPHP3, NPHP4 and ARL13B, indicating that CEP164 is likely to have other ciliary roles [1, 15, 24, 32].

Several studies have indicated a potential role for CEP164 in ATM/ATR-mediated DNA damage response (DDR) and UV-induced nucleotide excision repair pathways, however results are conflicting [14, 17, 33]. Likewise, data supporting a role for CEP164 in cell-cycle regulation is inconsistent [1, 2, 14, 15, 25]; both of these CEP164 functions however require further validation.

*CEP164* has numerous orthologs, including *M.musculus*, *D.melanogaster*, *C.reinhardtii and D.rerio* (**S1 Fig**). The murine ortholog of *CEP164* is a 30-exon gene, located on chromosome 9, qA5.2, which shares 77% identity to the full-length human *CEP164*. It encodes a 1333 amino acid protein, which shares 58% identity to the 1,460 amino acid human CEP164 protein. [34, 35]. Like human *CEP164*, murine *Cep164* may have multiple isoforms.

Given *CEP164* mutations cause NPHP-RC, we sought to determine the expression of CEP164 throughout human and murine development, particularly focusing on the cerebellar-retinal-renal phenotype. Collaboration with the Medical Research Council (MRC) Human Developmental Biology Resource (HDBR) allowed the procurement of human embryonic and foetal samples, which were compared to expression data from the murine 129/OlaHsd-*Cep164*$^{tm1a(EUCOMM)Wtsi/+}$ gene trap model.

## Materials and methods

### Study approval

The study was conducted with full ethical approval and consent. Ethical approval was obtained from the National Research Ethics Service Committee North East–Newcastle & North Tyne-side 1 (08/H0906/21+5). Human embryonic and foetal tissue samples were collected with appropriate consent and ethical approval, via the Medical Research Council (MRC) Wellcome trust-funded Human Developmental Biology Resource (HDBR). Informed and written consent was gained prior to the collection of control urine sample for isolation of human urine derived renal epithelial cells (hURECs). All methods were performed in accordance with the relevant ethical guidelines and regulations. Animal experiments were performed under Home Office Licences (United Kingdom) in accordance with the guidelines and regulations for the care and use of laboratory animals outlined by the Animals (Scientific Procedures) Act 1986. Protocols conducted were approved by the Animal Ethics Committee of Newcastle University and the Home Office, United Kingdom.

### Mouse genetics

C57BL/6NTac-*Cep164*$^{tm1a(EUCOMM)Wtsi/+}$ mice, generated for the International Mouse Phenotyping Consortium Initiative, were obtained from MRC Harwell [36]. These were back-crossed onto a 129/Ola-Hsd background, forming mice heterozygous for the gene trap, 129/OlaHsd-*Cep164*$^{tm1a(EUCOMM)Wtsi/+}$. 129/OlaHsd-*Cep164*$^{tm1a(EUCOMM)Wtsi/+}$ heterozygous (HET) mice were mated with wild type (WT) 129/OlaHsd-*Cep164*$^{+/+}$ mice, gaining pups HET: WT 1:1. Upon pre-mRNA processing of the *Cep164*$^{tm1a}$ allele, exon 3 of *Cep164* splices into the *LacZ/Neomycin* cassette, within intron 3, which introduces a premature termination codon and subsequent *Cep164* null allele (**S2 Fig**). The *LacZ* cassette contains an internal ribosomal entry site (IRES), allowing translation of a beta-galactosidase fusion gene upon activation of the native *Cep164* promoter, thus acting as a *Cep164* reporter gene. Upon addition of X-Gal substrate, beta galactosidase hydrolyses X-Gal forming 5-bromo-4-chloro-3-hydroxyindole-1, which oxidises to a blue precipitate, 5,5'-dibromo-4,4'-dichloro-indigo-2. HET 129/OlaHsd-*Cep164*$^{tm1a(EUCOMM)Wtsi/+}$ mice were used for X-Gal *Cep164* expression studies, as they are phenotypically normal, WT mice were used as littermate controls.

### *Cep164* genotyping

Ear or tail biopsies (embryos) were lysed for 1 h, at 95˚C, in 50 μl alkaline lysis buffer (25 mM NaOH, 0.2 mM EDTA, pH 12) and then neutralised in 50 μl neutralising buffer (40 mM trizma hydrochloride, pH 5–5.5). A PCR reaction using GoTaq G2 DNA Polymerase (Promega) for $Cep164^{tm1a(EUCOMM)Wtsi/+}$, was completed using the following primers flanking the second loxp site; F1 5' CTC CCA CAG TGA CAA ATG CC 3', R1 5' GGT AGT TGT TAC TTC TGT CAG 3' (Eurofins Genomics). Expected amplicon sizes are as follows; WT (141 bp), homozygous (Hom) (163 bp), HET (141 bp and 163 bp). PCR products were run on a 1.5% agarose gel, with GelRed Nucleic Acid GelStain (1:10,000) (Biotium) at 150V for 45 min. To confirm correct genotyping, representative samples were Sanger-sequenced (GATC-BIOTECH).

### Murine tissue collection, fixation, sectioning and staining

Murine tissue (kidney, brain, eye, heart, lung, liver, testes) was collected at P0.5/P1.5, P15.5 and P29.5/P30.5, using a standard dissection procedure. Tissues were fixed in 0.2% glutaraldehyde fix (0.2% glutaraldehyde, 2mM $MgCl_2$, 5mM EGTA in 1 x phosphate buffered saline PBS) for 90 min at 4˚C, washed in PBS and then stored in 15% sucrose in PBS overnight at 4˚C. Tissues were transferred to 30% sucrose in PBS and incubated at 4˚C overnight or until samples sunk. Tissues were frozen in OCT compound, and stored at -80˚C.

Sections of tissues were cut (10–12 μm) using a cryostat and mounted on charged glass slides. These were incubated in 0.2% glutaraldehyde fix (0.2% glutaraldehyde, 2mM $MgCl_2$, 5mM EGTA in PBS) for 5 min, washed in PBS for 10 min, and then washed in X-Gal wash (2 mM $MgCl_2$, 0.01% Sodium Deoxycholate, 0.02% NP-40 in PBS) for 10 min. Slides were incubated with X-Gal stain (1mg/ml of X-Gal DMSO in 2 mM $MgCl_2$, 0.01% Sodium Deoxycholate, 0.02% NP-40, 5mM Potassium Ferricyanide, 5mM Potassium Ferrocyanide in PBS) at 37˚C in the dark, until the blue precipitate stain intensity did not further increase or WT littermate controls showed endogenous beta galactosidase staining. Slides were washed in PBS and then dehydrated to 100% ethanol before clearing in Histoclear II (National Diagnostics) and mounting in DPX mounting medium (Sigma-Aldrich). Slides were imaged using the SCN400 Side Scanner (Leica).

### Murine embryo collection, wholemount fixation and staining

Murine embryos were collected at E9.5, E10.5 and E12.5 using a standard dissection procedure. Embryos were fixed in 0.2% glutaraldehyde solution (0.2% glutaraldehyde, 2mM $MgCl_2$, 5mM EGTA in PBS) for 1 hour on ice and washed with X-Gal wash (2 mM $MgCl_2$, 0.01% Sodium Deoxycholate, 0.02% NP-40 in PBS) prior to overnight storage at 4˚C. Embryos were incubated in X-Gal stain, consisting of 25mg/ml of X-Gal/DMSO solution in a 1 in 25 dilution of X-Gal staining buffer (2 mM $MgCl_2$, 0.01% Sodium Deoxycholate, 0.02% NP-40, 5mM Potassium Ferricyanide, 5mM Potassium Ferrocyanide in PBS) at 37˚C in the dark; incubation was completed once the blue precipitate stain intensity did not further increase or WT littermate controls started to show endogenous beta galactosidase staining. Embryos were washed in 1x PBS and dehydrated in 70% ethanol, prior to imaging using a Leica Man Stereomicroscope on the Axiovision software.

### Human tissue collection, fixation and processing

Human embryonic and foetal tissues were obtained from the MRC Wellcome Trust-funded Human Developmental Biology Resource (HDBR). CS23 whole embryos were fixed in 4% paraformaldehyde (PFA), for 72 hours, with an incision in the skull down the sagittal plane

and a slit in the abdomen at the umbilicus. Tissues were stored in methacarn prior to further processing. This protocol was used for processing of foetal eye, brain and kidney; the embryonic and foetal kidney were cut sagittally, to allow the fixative to penetrate. Human embryonic and foetal tissues were washed in increasing concentrations of ethanol, and then incubated in 3 changes of xylene before embedding in paraffin wax. Sections, on positively charged glass slides, were de-waxed in two washes of xylene, and then rinsed in two changes of absolute ethanol. Slides were incubated in methanol peroxide solution (0.5% $H_2O_2$) for 10 min to block endogenous peroxidase. Slides were rinsed in tap water and then antigen retrieval was completed using citrate buffer. After rinsing in Tris-buffered saline (TBS), slides were incubated with 10% goat serum (Vector) for 10 min at room temperature and then incubated with the primary rabbit anti-CEP164 (Human Protein Atlas, HPA37606) or rabbit anti-PAX6 (Covance, PRB-278-P-100) in goat serum with TBS overnight at 4˚C (**S1 Table**). After washing with TBS, slides were incubated with the goat anti rabbit BA-1000 biotinylated secondary antibody (Vector Laboratories) diluted in goat serum with TBS (1/500), for 30 min at room temperature. The secondary was washed with TBS and then slides were incubated with VECTASTAIN Elite ABC kit PK6100 tertiary complex (Vector Laboratories), for 30 min at room temperature. After TBS washes, the stain was developed for 10 min at room temperature using ImmPACT DAB peroxidase substrate (SK-6100) solution (Vector Laboratories). The slides were washed thoroughly in water, and counterstained with haematoxylin, dehydrated, cleared and mounted.

Due to the presence of endogenous biotin in kidney, an alternative two-step method was used for foetal kidney samples. Samples were incubated with a HRP goat anti-rabbit IgG peroxidase secondary (Vector) (1/500). After TBS washes, the stain was developed using ImmPACT DAB peroxidase substrate (SK-6100), as described above. No primary control sections were utilised as negative controls, anti-PAX6 was utilised as a positive control for the brain and cerebellum (**S1 Table**). Slides were imaged using the SCN400 Side Scanner (Leica).

### Immunofluorescence of human urine-derived renal epithelial cell

Human urine derived epithelial cells (hURECs) were isolated and cultured as described in [37]. Immunofluorescence staining with rabbit anti-CEP164 (Human protein atlas, HPA37606) and mouse anti-ARL3B (Proteintech, 66739-1-1g), were completed as per the protocol shown in [38, 39], however primary antibodies were incubated overnight at 4˚C. Cells were then mounted in Vectashield (Vector Laboratories Ltd, H-1200). Slides were imaged using the Zeiss axioimager.

### Image analysis

Images of human and murine tissues were analysed using the SCN400 Slide scanner software (Leica). Immunofluorescence staining of hURECs were imaged using the Zeiss axioimager, and processed using Zen Pro 2.3. Figs 1–4, and S3–S7 Figs were generated using Adobe Photoshop CS3 Extended.

## Results

### CEP164 expression during human and murine renal development

In the human embryonic and foetal kidney, immunohistochemical staining demonstrates that CEP164 is present in the metanephric epithelial-derived renal vesicles (**Fig 1A, 1B and 1C**), comma-shaped bodies (**Fig 1B**) and subsequent s-shaped bodies (**Fig 1C**), during all time points analysed (8PCW-18PCW). This expression pattern is maintained throughout nephrogenesis, with CEP164 present in the primitive nephron tubule (8PCW) (**Fig 1A**), but also in

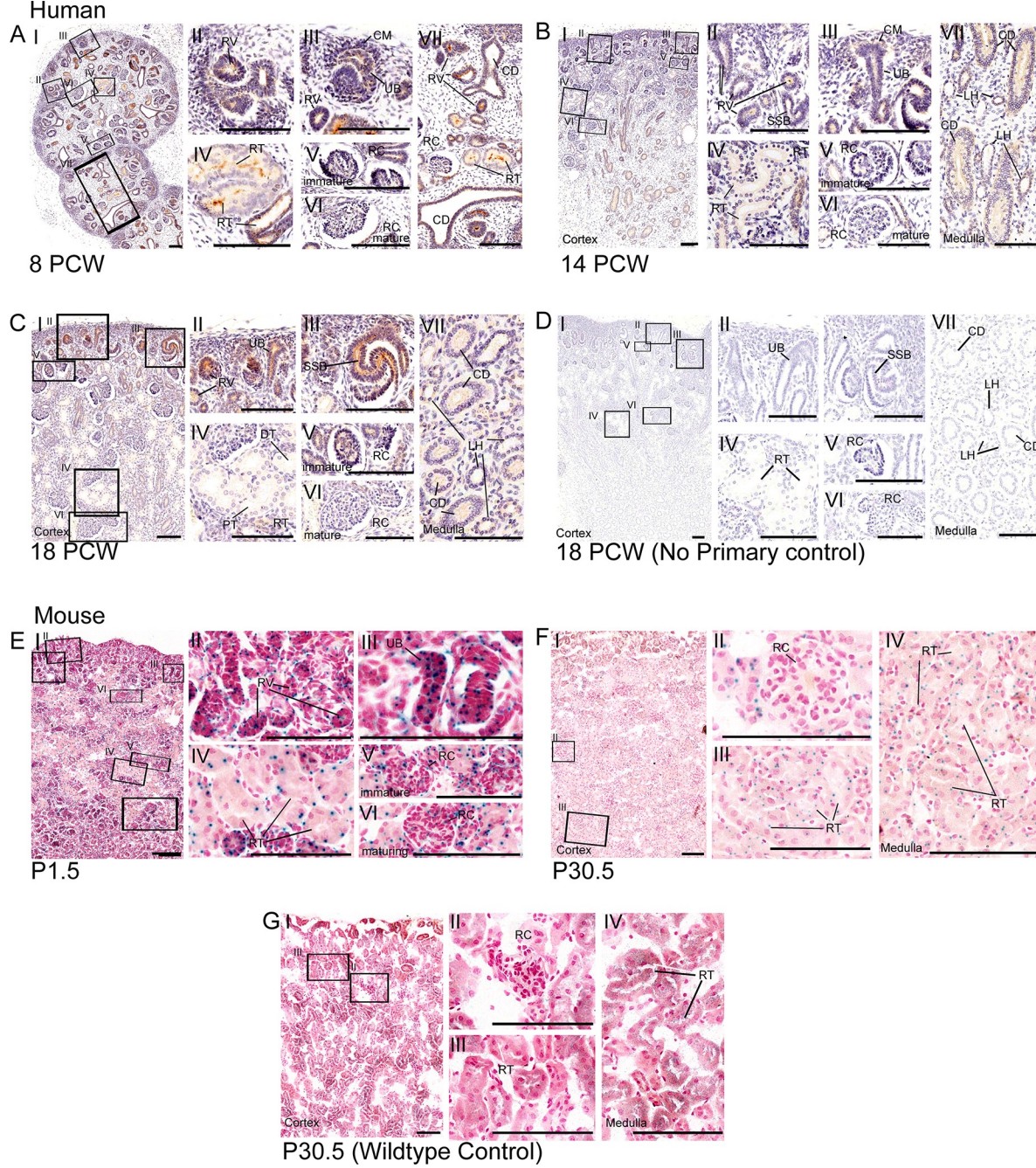

**Fig 1. Expression of CEP164 throughout human and murine renal development.** Human renal development (A-D), 8 PCW (A), 14 PCW (B), 18 PCW (C) respectively and 18 PCW no primary control (D). In the developing human kidney, CEP164 expression is seen in the apical membrane of the metanephric renal vesicles (A.II, B.II), s-shaped bodies (B.II, C.I, C.III) and developing renal tubules (A.IV.VII, B. IV.VII). From 14 PCW CEP164 expression is seen at the apical membrane of all developing tubular segments including the distal and proximal tubules (B.IV, C.IV) and loop of Henle segments (B.VII, C.VII). CEP164 expression is seen in the cells of the ureteric bud (A.III, B. III, C.II), and the subsequent collecting duct at both the apical and basal membrane (A.VII, B.VII, C.VII). CEP164 expression is seen in the glomerulus of the developing immature renal corpuscle (A.V, B.V, C.V), and weakly in the matured renal corpuscle (A.VI, B.VI, C.VI). No primary controls demonstrate no background DAB staining, as represented by 18 PCW time-point (D.I-VII). Murine postnatal renal development (E-G), P1.5 (E), P30.5 (F) and P30.5 WT control (G). In the murine kidney *Cep164* expression is present in the developing renal vesicles (E.I.II) and ubiquitously throughout subsequent nephron renal tubules (E.IV, F.I.III.IV). *Cep164* expression is seen in the glomerulus of the developing renal corpuscle (E.V), but expression seems to be lost with maturity (E.VI, F.II). No endogenous beta galactosidase staining is present in the murine kidney, as shown by the representative image WT control at P30.5 (G). All scale bars represent 100 µm. Cap mesenchyme (CM), collecting duct (CD), distal tubule (DT), loop of Henle (LH), postnatal day (P), post conception weeks (PCW), proximal tubule (PT), renal corpuscle (RC), renal tubule (RT), renal vesicle (RV), s-shaped body (SSB), ureteric bud (UB).

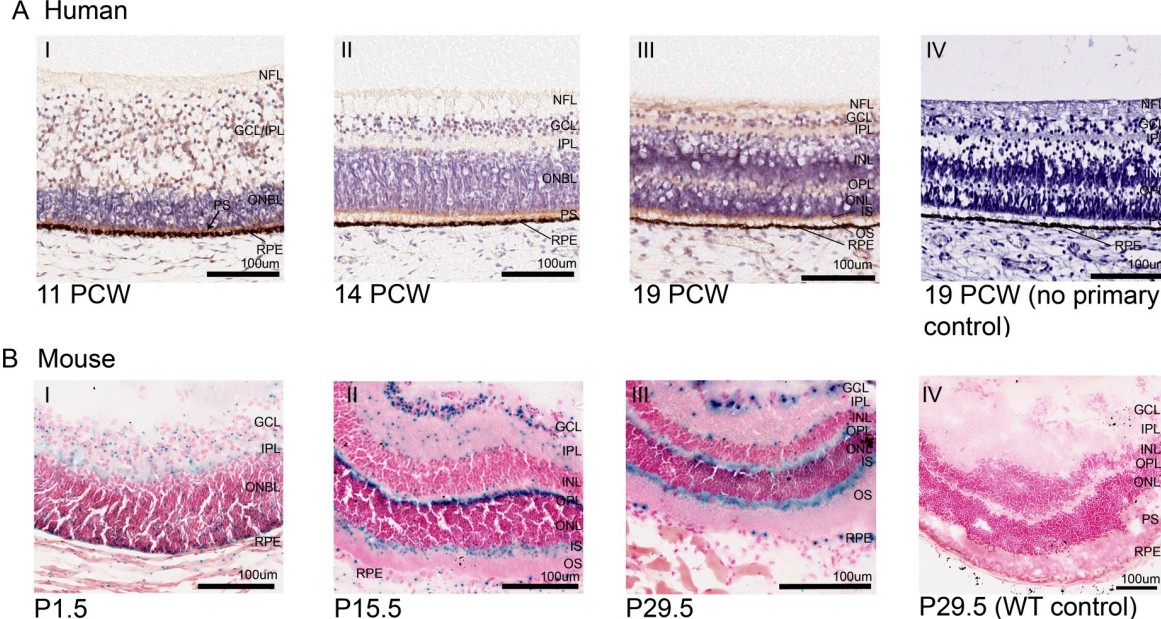

**Fig 2. CEP164 expression in the developing human and murine retina.** Human Retina (A), 11 PCW (A.I), 14 PCW (A.II), 19 PCW (A. III) and 19 PCW no primary control (A.IV). In the developing human retina, weak CEP164 expression is seen in the nerve fibre layer (NFL) and ganglion cell layer/inner plexiform layer (GCL/IPL) (A.I) and strong expression in outer neuroblastic (ONBL) photoreceptor precursors (black arrow) (A.I). By 14 PCW, CEP164 expression is seen in the developed inner plexiform layer (IPL) and the developing photoreceptor layers (A.II). At 19 PCW, CEP164 expression is seen in the nerve fibre layer (NFL), inner plexiform layer (IPL), outer plexiform layer (OPL) and photoreceptor layer, with enhancement in the inner photoreceptor segments (IS) (A.III). No background staining is present in the no primary controls (A.IV). Murine Retina (B), P1.5 (B.I), P15.5 (B.II), P29.5 (B.III) and P29.5 WT control (B. IV). In the developing murine retina, *Cep164* expression is seen in the inner plexiform layer (IPL), ganglion cell layer (GCL) and outer neuroblastic layer (ONBL) (B.I). At P15.5, *Cep164* expression is seen in the ganglion cell layer (GCL), the outer plexiform layer (OPL) and inner segment (IS) of the photoreceptor layer (B.II). There is also punctate expression in the inner plexiform layer (IPL) and edges of the nuclear cell layers (B.II). Retinal pigment epithelium (RPE) also shows *Cep164* expression (B.I, B.II). This murine retinal expression patterning is maintained once retinal layers have been formed (B.III). Controls demonstrate no endogenous beta galactosidase activity in the murine retina (B.IV). Ganglion cell layer (GCL), inner nuclear layer (INL), inner plexiform layer (IPL), inner segment (IS), nerve fibre layer (NFL), outer neuroblastic cell layer (ONBL), outer nuclear layer (ONL), outer plexiform layer (OPL), outer segment (OS), photoreceptor segment layer (PS), retinal pigment epithelium (RPE).

more defined proximal and distal nephron segments, as well as cells of the loop of Henle, as seen from 14 PCW (**Fig 1B and 1C**). Specifically, CEP164 expression is enriched at the apical membrane of epithelial cells lining the developing nephron lumen, during all developmental time points analysed (8PCW- 18PCW) (**S3 Fig**). CEP164 does not appear to be expressed, or is expressed very weakly, in the metanephric cap mesenchyme (**Fig 1A, 1B and 1C**).

In the developing human kidney, CEP164 is expressed in the glomerulus of the renal corpuscle at 8 PCW-18 PCW, indicating podocyte CEP164 expression (**Fig 1A, 1B and 1C**). The expression data indicates that CEP164 expression is reduced as glomeruli mature; this is most clearly seen at 18 PCW (**C.V.VI**).

The human embryonic/foetal ureteric bud, which forms the collecting duct, is derived from the metanephrogenic diverticulum. Both the ureteric bud and collecting duct also show CEP164 expression in the apical membrane of the epithelial cells lining the tubular lumen (8PCW-18PCW) (**Fig 1A, 1B and 1C**). CEP164 is also expressed at the basolateral membrane of the collecting duct tubule (8PCW-18PCW). CEP164 expression is not present in the human renal interstitium (**Fig 1A, 1B and 1C**). Lack of background DAB staining in no primary controls indicate that CEP164 expression in the human kidney was specific (**Fig 1D** and **S4 Fig**).

In the murine kidney (129/OlaHsd-*Cep164*$^{tm1a(EUCOMM)Wtsi/+}$), the *LacZ* reporter assay demonstrates that *Cep164* expression correlates with human kidney CEP164 expression.

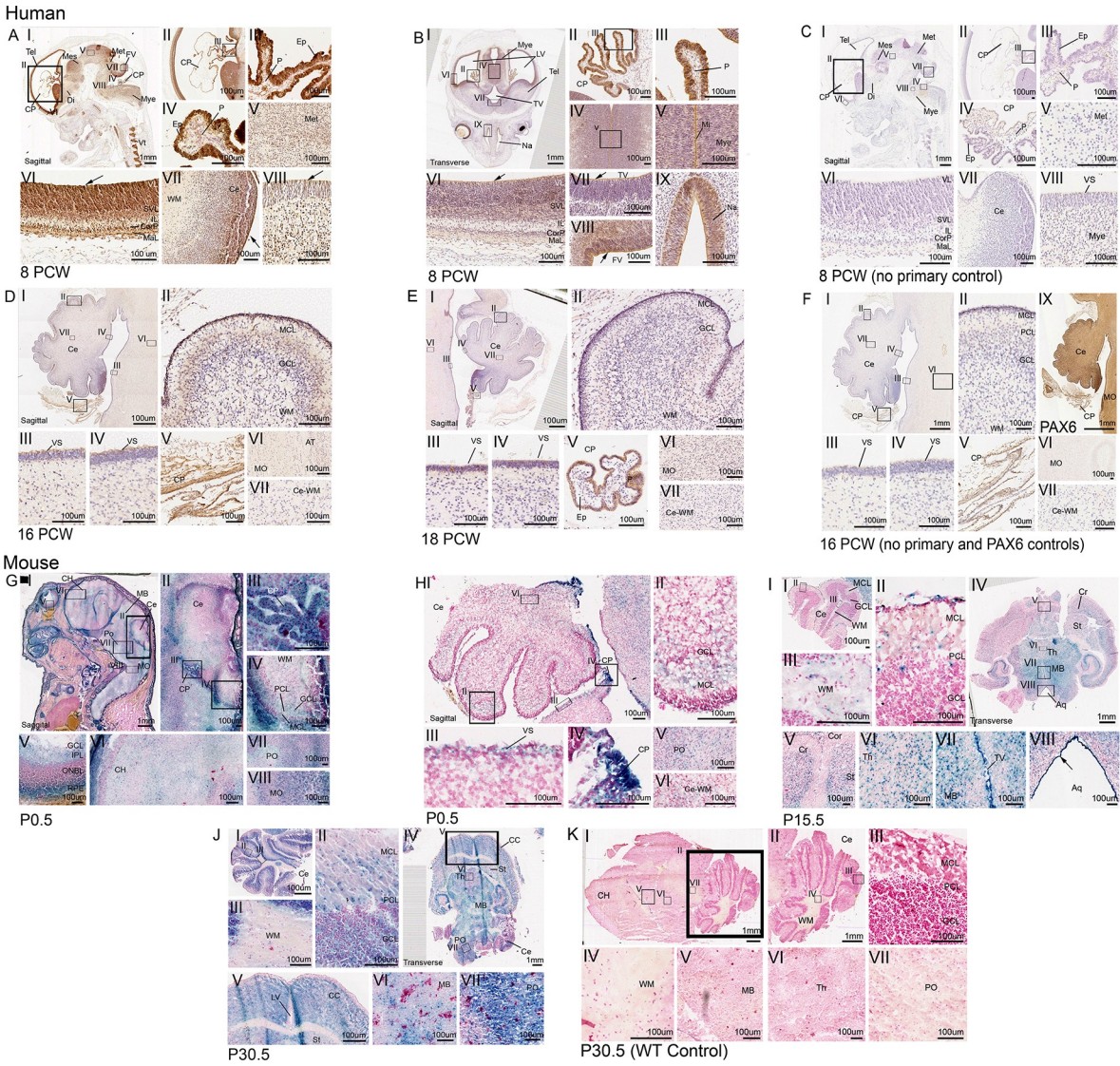

**Fig 3. CEP164 expression in the developing human and murine brain, focusing on the hindbrain.** Developing human brain (A-F), 8 PCW (A-B), 8 PCW no primary control (C), 16 PCW hindbrain (D),18 PCW hindbrain (E) and 16 PCW hindbrain no primary control (F). In the developing human brain at 8 PCW, CEP164 expression is seen in the neuroepithelium surrounding the telencephalon (A.I.II.VI, B.I. VI), diencephalon (A.I), mesencephalon (A.I), metencephalon (A.I.V) and myencephalon (A.I.VIII, B.I.IV.V), with defined expression in apical neuroepithelium cells lining the brain ventricles (A.I.VI.VIII, B.I.VI.VII.VIII) demonstrated by black arrows. Defined expression is present in the layers of the telencephalon (A.VI, B.VI). There is defined expression of CEP164 in the human brain midline (B.IV.V) and the nasal epithelium (B.IX). At 8PCW, the choroid plexus shows strong CEP164 expression in the ependymal cells (A.II.III.IV, B.II.III), with seemingly weaker expression in the choroid plexus pia matter (A.IV, B.II.III). The ependymal expression is maintained at 16 PCW and 18 PCW (D.I.V, E.I.V). CEP164 expression is seen in the migrating molecular cell layer of the developing cerebellum (D.I.II), which seems to be lost with molecular cell layer maturation (E.I.II). In the human hindbrain the apical membrane of the pons and the cerebellum demonstrate defined CEP164 expression (D.III,IV), which appears to be lost with maturation (E.III.IV). Weak CEP164 expression is seen in the human medulla oblongata (D.VI, E.VI) but not in the white matter (D.VII, E.VII). Developing murine brain (G-K), P0.5 brain (G) P0.5 hindbrain (H), P15.5 (I),P30.5 (J) and P30.5 WT control (K).*Cep164* expression is widespread throughout the developing murine brain (P0.5-P30.5) (G.I, I.IV, J.IV) with expression in the cortex, striatum and cerebrum of the maturing cerebral cortex (G.I.VI, I.IV.V, J.IV.V) expression in the midbrain (G.I, I.IV.VII, J.IV.VI) and thalamus (I.IV.VII). *Cep164* expression is strong and defined in the ventricular neuroepithelium (I.VII.VIII). The murine choroid plexus demonstrates strong *Cep164* expression, however it is not defined to a specific cell type (G.III, H.IV). In the murine hindbrain, at P0.5, *Cep164* is expressed in the migrating molecular layer of the cerebellum (G.II.IV, HI.II), this is maintained with expression also in the molecular layer, ganglion cell layer and Purkinje cell layer at P15.5 and P30.5(I.I.II,J.I.II). *Cep164* expression is seen in the in the murine pons (G.VII, H.V, J. VII) and medulla oblongata (G.VIII), and weakly in the cerebellar white matter (G.IV, H.VI, I.III), which seems to be lost with maturation (J.III). The representative P30.5 control demonstrates no endogenous beta galactosidase staining (K). Aquaduct (Aq), axon tract (AT), cerebellum (Ce), cerebral cortex (CC), cerebral hemisphere (CH), cerebrum (Cr), choroid plexus (CP), cortex (Cor), cortical plate (CorP), diencephalon (Di), ependymal cells (Ep), fourth ventricle (FV), ganglion cell layer (GCL), inner plexiform layer (IPL), intermediate zone (IL), lateral ventricle (LV), marginal layer (MaL), medulla oblongata (MO), mesencephalon (Mes), metencephalon (Met), midbrain (MB), midline (Mi), molecular cell layer (MCL), myencephalon (Mye), nasal

epithelium (Na), outer neuroblastic layer (ONBL), pia (P), pons (PO), purkinje cell layer (PCL), retinal pigment epithelium (RPE), striatum (St), sub-ventricular layer (SVL), telencephalon (Tel), thalamus (Th), third ventricle (TV), ventricular layer (VL), ventricular surface (VS), white matter (WM).

Correspondingly, *Cep164* is expressed in the developing murine renal vesicles, s-shaped bodies (P1.5) (**Fig 1E**) and the subsequent renal tubules, including nephron segments in both the cortex and medulla (P30.5) (**Fig 1E and 1F**). The staining indicates that there is potentially cell-specific expression within each of the tubular segments. *Cep164* is also expressed in the glomeruli of the renal corpuscle at P1.5, which is not present in mature glomeruli at P30.5 (**Fig 1E and 1F**). Likewise, *Cep164* is expressed in the developing mesonephric ureteric bud (**Fig 1E**). Wildtype controls demonstrate that there is very low endogenous beta galactosidase expression in the kidney (**Fig 1G** and **S5 Fig**).

## CEP164 expression during human and murine retinal development

In the developing human retina, CEP164 expression is widespread yet defined (**Fig 2A**). In early retinal development (11 PCW) CEP164 is expressed weakly in the developing nerve fibre layer (NFL) (fibrous extensions from the optic nerve) and nerve fibres in the differentiating ganglion cell layer (GCL)/inner plexiform layer (IPL) (**Fig 2A**). The basally located layer of cone precursors, which has differentiated from the outer neuroblastic cell layer (ONBL), also demonstrates strong CEP164 expression (**Fig 2A**). By 14 PCW, CEP164 expression is defined to the NFL, the IPL and the photoreceptor layer which now has both developing rods and cones (**Fig 2A**). Later in development, 19 PCW, once all of the primitive retinal layers have formed, CEP164 expression is maintained in the nerve fibres of the NFL, the ganglionic and inner nuclear nerve fibril synapses of the IPL, and the inner segments (IS) and outer segments (OS) of the photoreceptor cell layer (**Fig 2A**). At 19 PCW, CEP164 expression is also present in the developing outer plexiform layer (OPL) which contains nerve fibril synapses between the inner nuclear layer (INL) and the outer nuclear layer (ONL). CEP164 expression in the retinal pigment epithelial layer (RPE) cannot be defined due to its natural dark colouring, although no endogenous background staining is present in the other retinal layers (**Fig 2A** and **S4 Fig**).

Murine *Cep164* retinal expression in 129/OlaHsd-*Cep164*$^{tm1a(EUCOMM)Wtsi/+}$ largely corresponds to the human CEP164 expression patterns, with expression maintained in the developing IPL, OPL and PS layer (P1.5-P29.5) (**Fig 2B**). However, there are some clear differences. *Cep164* is expressed in retinal precursor cells throughout the murine ONBL, which in the developing human retina is restricted to the cone precursor cells (**Fig 2B**). Additionally, strong *Cep164* expression is present and maintained throughout the GCL (P1.5-P29.5) (**Fig 2B**). *Cep164* is also expressed sparsely in some cells of the INL and ONL, however these are at the boundaries of the plexiform layers (P15.5-P29.5) (**Fig 2B**). Notably, at later stages of development, *Cep164* expression is clearly defined to the IS of the photoreceptor cell layer (P15.5-P29.5) (**Fig 2B**). Additionally, the RPE demonstrates strong, cell-specific *Cep164* expression throughout development (P1.5-P29.5) (**Fig 2B**). WT controls demonstrate no endogenous X-Gal staining (**Fig 2B** and **S5 Fig**).

## CEP164 expression during human and murine neuronal and cerebellar development

CEP164 is expressed widely throughout human brain development (8PCW-18PCW) (**Fig 3**). At 8PCW CEP164 is strongly expressed in the neuroepithelium of the developing telencephalon (Tel) (**Fig 3A and 3B**), which is defined in cells lining the lateral ventricle (LV), including

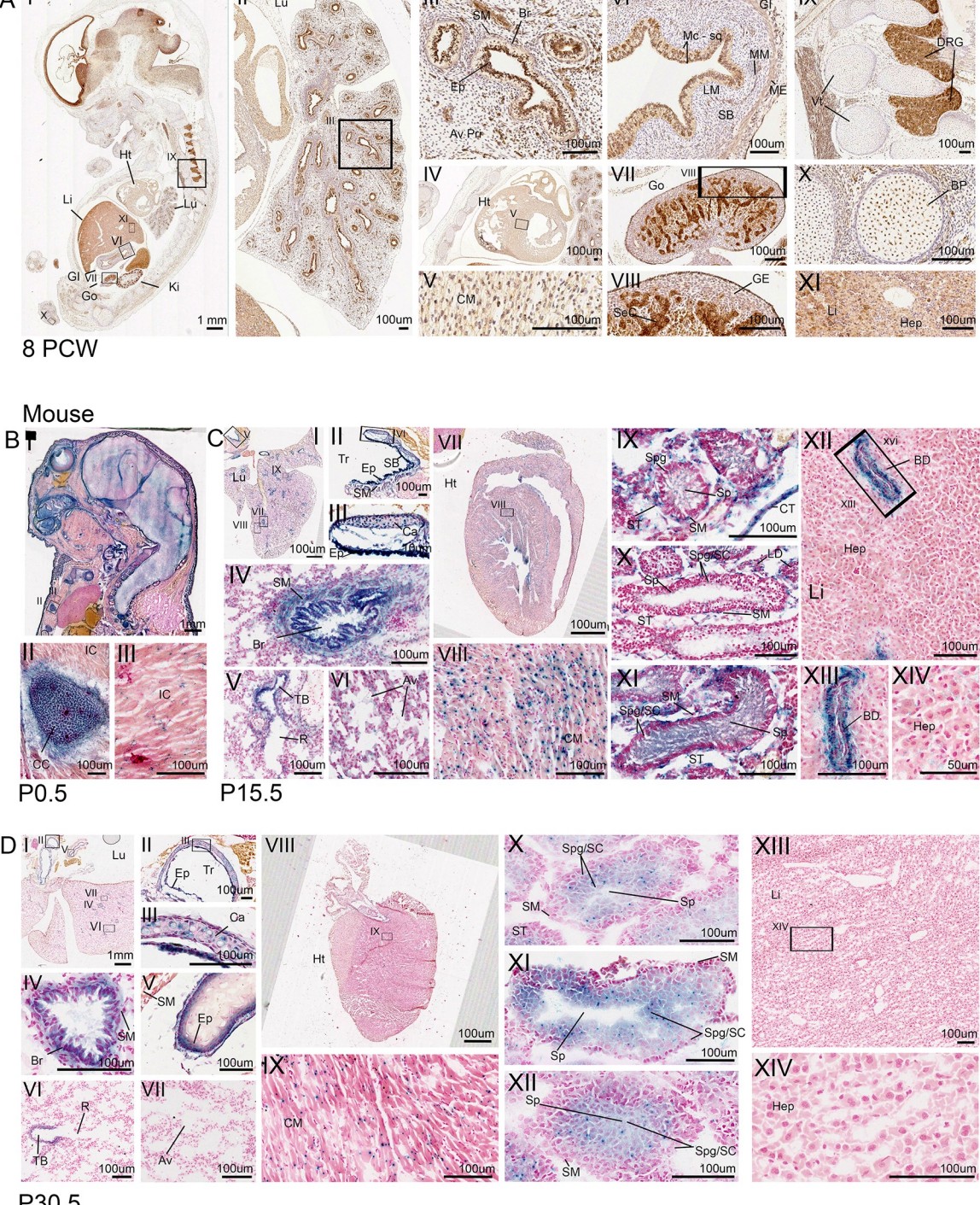

**Fig 4. CEP164 expression in secondary organs throughout human and murine development.** Human 8 PCW (A). Murine (B-D), P0.5 (B), P15.5 (C), P30.5 (D). In the developing human, lung CEP164 expression is seen in the respiratory epithelial lining of the bronchi and bronchioles, with weaker expression in the smooth muscle and alveoli (AI.II.III). In the developing murine lung, *Cep164* is expressed in the respiratory epithelial lining of the bronchi and bronchioles (C.I.IV, D.I.IV), with additional *Cep164* expression seen in the respiratory epithelial cells lining the trachea and the tertiary bronchioles (C.I.II.III.V, D.I.II.III.V.VI). *Cep164* expression is seen in murine alveoli at P15.5 (C.VI), which seems to be lost with maturity (P30.5) (D.VII). *Cep164* expression is seen within the cartilage of the trachea (C.II.III, D.II.III). In both the human (A.IV.V) and murine hearts (C.VII.VIII, D.VIII.IX), expression is seen the developing cardiomyocytes. In the human gastrointestinal tract CEP164 expression is seen in the inner mucosae squamous epithelial cell layer,

muscularis mucosae cell layer and the external muscularis cell layer (A.VI). In the developing human gonads, CEP164 expression is seen in the germline epithelium and seminiferous cord (A.VII.VIII). In the developing murine testes, at P15.5, *Cep164* expression is seen in the seminiferous tubules, specifically the smooth muscle cells spermatogonia, spermatocytes and most strongly in spermatids (C.IX.X. XI); expression in leydig cells and connective tissue can also be seen (C.X). At P30.5, Cep164 expression is defined to the spermatogonia, spermatocytes and spermatids (D.X.XI.XII). CEP164 expression is seen in the dorsal root ganglia of the human spinal cord, (A.I.IX) with weaker expression also present in the vertebrae primordia (A.I.IX) and bone primordia (A.I.X). *Cep164* expression is seen in the murine developing costal cartilage (B.I.II) with weaker expression in intercostal muscle (B.I.III). The human foetal liver demonstrates some evidence of CEP164 expression (A.XI), however this is undetermined due to high endogenous peroxidase activity in the liver. The developing murine liver shows *Cep164* expression in epithelial cells lining the hepatic portal veins at P15.5 (C.XII.XIII), this is not seen by P30.5. There is no expression in hepatocytes (C.XIV, D.XIV). Aveoli (Av), alveoli primordia (Av Pri), bile duct (BD), bone primordia (BP), bronchiole (Br), cardiomyocyte (CM), cartilage (Ca), connective tissue (CT), costal cartilage (CC), dorsal root ganglia (DRG), epithelial (Ep), gastrointestinal tract (GI), germline epithelium (GE), gonad (Go), heart (Ht), hepatocytes (Hep), intercostal muscle (IC), kidney (Ki), lamina propria (LM), liver (Li), lung (Lu), muscularis externa (ME), muscularis mucosae (MM), respiratory bronchiole (R), seminiferous cord (SeC), seminiferous tubule (ST), smooth muscle (SM), spermatids (Sp), spermatocytes (SC), spermatogonia (Spg), squamous cell mucosae (Mc-sq), submucosae (SB), trachea (Tr), terminal bronchi (TB), vertebrae (Vt).

the ventricular layer (VL), subventricular layer (SVL), cortical plate (CorP) and marginal layer (MaL), (**Fig 3A and 3B**). At 8 PCW, CEP164 is also present in the developing neuroepithelium of the diencephalon (Di) (**Fig 3A and 3B.**)**,** mesencephalon (Mes) (**Fig 3A.I**), metencephalon (Met) (**Fig 3A**) and myencephalon (Mye) (**Fig 3A, B.I.IV.V**), with expression strongest in the apical epithelium lining the brain ventricles as highlighted by a black arrow (**Fig 3A and 3B**). At 8PCW there is strong, well-defined CEP164 expression in the midline of the myencephalon (**Fig 3B**), as well as cells of the nasal epithelium (**Fig 3B**). Throughout human brain development (8PCW-18PCW) the choroid plexus demonstrates strong CEP164 expression, specifically in the ependymal cells, and more weakly in the choroid plexus pia matter (**Fig 3A, 3B, 3D and 3E**). The developing human cerebellar folds show defined CEP164 expression in the migrating molecular cell layer at 16 PCW (**Fig 3D**), but this seems to be reduced by 18PCW (**Fig 3E**). CEP164 is expressed at the apical membrane of the ventricular surface of the pons and cerebellum (**Fig 3A, 3D and 3E**) as well as the myencephalon, specifically in the axon tracts of the medulla oblongata (**Fig 3D and 3E**).This expression pattern can be seen from 8PCW to 18 PCW, however CEP164 expression levels seem much weaker by 18 PCW. At 8PCW CEP164 is expressed in the white matter (**Fig 3A**), but this appears to be lost with maturation of the cerebellum (**Fig 3D and 3E**). Notably the human foetal brain negative controls demonstrate very low background staining, specifically in the choroid plexus region (**Fig 3C, 3F** and **S4D–S4F Fig**). A PAX6 antibody was used as a positive control (**Fig 3F and S4 Fig**).

Corresponding with the developing human brain, *Cep164* is expressed in the neuroepithelium of the murine telencephalon (129/OlaHsd-*Cep164*$^{tm1a(EUCOMM)Wtsi/+}$) (cerebral hemisphere) (**Fig 3G**), diencephalon (**Fig 3G**), mesencephalon (midbrain) (**Fig 3G**), metencephalon (pons) (**Fig 3G**) and myencephalon (medulla oblongata) at P0.5 (**Fig 3G**). At later stages of development (P15.5-P30.5), clear *Cep164* expression can be seen throughout the brain including the cerebrum and striatum of the cerebral cortex (**Fig 3I and 3J**), the thalamus of the diencephalon (**Fig 3I and 3J**), the midbrain (**Fig 3I and 3J**) and hindbrain pons and medulla oblongata (**Fig 3J**). Additionally, correlating with human foetal expression, *Cep164* is expressed strongly in neuroepithelium lining the brain ventricles, including the third ventricle, fourth ventricle and cerebral aquaduct (**Fig 3I**).

At P0.5 to P15.5 *Cep164* is expressed in the migrating molecular cell layer and Purkinje layer of the cerebellum (**Fig 3H and 3I**)**.** Later in development, *Cep164* is also expressed in the ganglion cell layer (P30.5) (**Fig 3J**). *Cep164* is also expressed at the apical membrane of the ventricular surface of the cerebellum at P0.5 and P15.5 (**Fig 3H**). Unlike the human cerebellar expression, *Cep164* is expressed in the white matter of the cerebellum which is present, although seems to be reduced by P30.5 (**Fig 3G, 3H, 3I and 3J**). *Cep164* is also expressed in the

ependymal cells of the choroid plexus, which is demonstrated in P0.5 sections (Fig 3G and 3H). Murine controls demonstrate that there is no endogenous beta galactosidase expression in the murine brain, apart from low endogenous staining in the choroid plexus at P0.5 (Fig 3K and S5 Fig).

### CEP164 expression during human and murine development in other tissues

CEP164 is expressed widely throughout the developing human embryo at 8PCW (Fig 4A.). In the 8PCW lung, CEP164 is expressed most strongly in the epithelial cells lining the lumen of the bronchi and bronchioles (Fig 4A). Weaker CEP164 expression is also present in the bronchiole smooth muscle, and the alveoli primordia (Fig 4A). In the 8PCW developing heart, CEP164 is expressed in cardiomyocytes (Fig 4A). The gastrointestinal tract also shows strong defined CEP164 expression in the inner mucosa squamous epithelium cell layer, muscularis mucosae cell layer and the external muscularis externa layer at 8PCW (Fig 4A). At 8PCW in the developing gonads, CEP164 is expressed weakly in the germline epithelium, but strongly in the seminiferous cord tubules (Fig 4A). Additionally, CEP164 is expressed strongly in the dorsal root ganglia of the spinal cord, with weak expression in the developing vertebrae (Fig 4A). CEP164 is also expressed in developing bone primordia of the limbs at 8 PCW (Fig 4A). Due to endogenous beta galactosidase activity the liver (S4 Fig) it cannot be determined whether CEP164 is expressed in human embryonic hepatocytes. The other tissues show no endogenous beta galactosidase staining (S4 Fig).

Murine postnatal tissues also demonstrate widespread *Cep164* expression. At P0.5 *Cep164* is expressed in the developing costal cartilage primordia and intercostal muscles (Fig 4B). Correlating with the human lung, in the developing murine P15.5 and P30.5 lung, *Cep164* is present in the epithelial cells lining the trachea, bronchioles and terminal/respiratory bronchioles (Fig 4C and 4D), as well as the smooth muscle and cartilage of the trachea and bronchi (Fig 4C and 4D) Weak *Cep164* expression is seen in the alveoli at P15.5 (Fig 4C), which seems to be reduced by P29.5 (Fig 4D). Like human embryonic tissues, *Cep164* is present in cardiomyocytes at P15.5 and P30.5 (Fig 4C and 4D). In postnatal P15.5 murine testes, *Cep164* is expressed in the connecting tubules and Leydig cells (Fig 4C and S7 Fig). In the seminiferous tubules, *Cep164* is expressed in the smooth muscle cells and weakly in developing spermatogonia and spermatocytes (Fig 4C and S7 Fig). Strong *Cep164* expression is present in the spermatid tails (Fig 4C and S7 Fig). By P30.5 *Cep164* expression is defined to developing spermatogonia, spermatocytes, and spermatids (Fig 4D and S5 Fig). At P15.5 *Cep164* is expressed strongly in cells lining the ciliated cholangiocyte cells of the bile duct, however hepatocytes do not show any expression (Fig 4C). There is no *Cep164* expression present in the murine liver at P30.5. X-gal staining of WT tissues, indicate that there is no endogenous beta galactosidase activity in the costal cartilage and intercostal muscles, cardiomyocytes, testes) and hepatocytes (S5 Fig). There is low endogenous beta galactosidase activity in the bronchioles at P15.5, but not at P30.5 (S5 Fig).

Wholemount *Cep164* expression studies (E9.5, E10.5, E12.5), demonstrate that *Cep164* expression is widespread throughout murine embryonic development, including the bronchial arches and organ primordia, as well as the developing CNS, heart and limbs (S6 Fig). Taken together this indicates that CEP164 expression is widespread throughout human and murine development in tissues beyond the cerebello-retinal-renal structures associated with typical disease phenotypes.

### Discussion

In this study we have described the expression of CEP164 in the developing human embryo and foetus, utilising immunohistochemistry, focusing upon the kidney, retina and cerebellum.

Conservation of CEP164 expression was explored using a *LacZ* gene trap assay to characterise *Cep164* expression in corresponding murine tissues (129/OlaHsd-*Cep164*$^{tm1a(EUCOMM)Wtsi/+}$). Notably, during murine development, the kidney, retina and cerebellum continue to develop postnatally. This reflects the chosen murine postnatal timepoints in this study, corresponding to estimated human embryonic and foetal developmental timepoints (**S2**–**S4 Tables**) [40–57].

Our results demonstrate that during human and murine development CEP164 is expressed widely, in multiple organs. Notably, CEP164 expression is clearly defined within tissues. In the developing human kidney, CEP164 is expressed in the apical epithelium membrane of meta-nephric-derived renal vesicles and subsequent nephron tubules (8PCW-18PCW), correlating with the presence of primary cilia [58]. Primary cilia are vital in the kidney for mechanosensation and chemical signal transduction, which is required for orientated cellular divisions during development and kidney maintenance. As CEP164 expression seems to be low/not present in the cap mesenchyme, it could be speculated that CEP164 expression is switched on during mesenchymal-epithelial transition, which coincides with the formation of the primary cilium during establishment of apical-basal polarity, cell junctions and lumen formation [58]. CEP164 is expressed within the glomerulus of the renal corpuscle, and there appears to be a reduction in CEP164 expression with maturity. Interestingly, this correlates with the loss of primary cilia in podocytes seen with glomeruli maturity in rats [59]. It could be postulated that in the developing human kidney, CEP164 expression is lost in maturing glomeruli due to a similar mechanism [59]. This could protect against overstimulation of calcium-mediated signalling due to an increase in glomerular filtration rate with development [59]. CEP164 is also expressed in the ureteric bud, which develops into the primary ciliated collecting duct network (8PCW-18PCW), consistent with numerous IMCD3 expression studies [1, 19, 20, 25, 32]. The human CEP164 expression pattern is conserved in the murine kidney (P0.5-P29.5); there may be cell-specific *Cep164* expression within nephron tubular segments, however this needs to be further studied.

Although our data is focused upon development, the human protein atlas indicates that CEP164 expression is maintained in the adult kidney [60]. A combination of P30.5 expression data and previous wholemount *Cep164* expression studies also indicate that *Cep164* expression is maintained in the adult mouse kidney [36]. Together this suggests that CEP164 may have roles in both normal renal development and maintenance of kidney function in the human and mouse. Aberrant CEP164 activity in the kidney could contribute to abnormal primary ciliary function, causing dysregulation of cell division and cell signalling, leading to cystogenesis, which has been suggested in a recent study using murine *Cep164* collecting duct cell knockdown [61]. This is consistent with the NPHP-RC phenotype present in most patients with *CEP164* mutations [1].

Retinal photoreceptor cells have a specialised connecting primary cilium between the inner and outer segment of the photoreceptors. It therefore seems reasonable that CEP164, a distal appendage centrosomal protein, is expressed in photoreceptor segment in the developing human and murine retina. CEP164 expression has been previously identified in the murine connecting cilium, further confirming our results [1, 62]. In the murine retina, C*ep164* expression is defined to the inner segment of the photoreceptor layer by P15.5. This correlates with previous preliminary *Cep164 In Situ* hybridisation studies [63]. Combining human and murine data, it can be suggested that the *Cep164* is transcribed and translated in the inner segment but then the CEP164 protein is transported to the outer segment of the photoreceptor cells. It can be hypothesised that with abnormal functioning/loss of CEP164, there could be atypical outer segment formation, which could lead to the accumulation of phototransduction proteins that could trigger cell loss, leading to a retinal degeneration phenotype, as seen in some *CEP164* NPHP-RC patients [1, 9, 47].

Multiple studies have suggested that primary cilia are present and functional in neurones [64]. This could explain why CEP164, a cilial protein, is expressed in other retinal layers. It is also well established that CEP164 is present at the basal body of RPE cells, clarifying the RPE expression results from our study [1, 2, 15, 17, 23].

Primary cilia are vital in the developing brain for sonic hedgehog (SHH) signalling, which is needed for proliferation of neuronal granular cell precursors in vertebrates, being a major driver of cerebellar precursors [45, 46]. SHH is also required for the cell-specific expansion of postnatal progenitors [64]. Wnt signalling, transduced by the primary cilia, is also required for neuronal patterning, cell proliferation and neuronal migration [46–48]. In the adult nervous system, primary cilia are thought to be required in stem cell regulation and tissue regeneration, however the full extent of primary cilia function in the adult central nervous system is still to be determined [64, 65]. With widespread expression of the cilia protein CEP164 in the developing human and murine brain, it can be suggested that CEP164 has a functional role in human and mouse brain development. Potentially, aberrant function of CEP164, may lead to brain dysplasia via abnormal primary cilia functioning in neuronal precursors. Some CEP164 NPHP-RC patients show neurological phenotypes, including abnormal developmental delay, intellectual disability and in one patient, cerebellar vermis aplasia, an archetypal feature of Joubert syndrome. Another patient also experiences seizures [1, 9].

Leptin receptors are situated in or near the primary cilia of the choroid plexus, and in the hypothalamus [66]. Leptin signalling is vital in the hypothalamic satiety pathway. Patients with mutations in centrosomal BBS proteins demonstrate hyperleptinemia due to leptin resistance, which causes hyperphagia and subsequent weight gain. It could be hypothesised that CEP164, which is expressed in the developing choroid plexus of the human (8 PCW, 16 PCW, 18PCW) and murine brain (P0.5), may localise to the choroid plexus primary cilium, and be indirectly involved in leptin receptor localisation or functioning. This could contribute to the obesity phenotype seen in some CEP164 NPHP-RC patients [1, 9].

Ependymal cells of the choroid plexus, which contain motile cilia, are required for the production and regulation of cerebral spinal fluid (CSF). Additionally, the lateral ventricle epithelium contains motile cilia which is important for the movement of CSF throughout the brain ventricles. Previous studies have established a ciliogenesis role for CEP164 in the motile cilium and demonstrated CSF defects including hydrocephalus in zebrafish and murine models lacking functional CEP164 [24]. This further validates the strong CEP164 expression in ependymal cells and the neuroepithelium lining the brain ventricles.

CEP164 is also expressed in tissues secondary to the cerebellar-retinal-renal phenotype, some of which have motile cilium. Taking together results from a previous murine CEP164 motile cilia study [24], it is plausible that CEP164 is expressed in ciliated epithelial cells, including those of the trachea, bronchi and bronchioles in the developing human (8 PCW) and murine (P15.5-P29.5) lung. Previous studies indicate that this is likely to be maintained into adulthood [36, 60]. *Cep164* is also expressed in murine spermatids, which have a flagellum, effectively a modified motile cilium [67]. Studies have demonstrated that CEP164 is expressed in the capitulum and striated columns of the human sperm neck [68]. Using information from a recent FOXJ1 mediated knockout of *Cep164* and this expression data, it could be hypothesised that CEP164 is required for the formation, maintenance and functioning of motile cilia within the respiratory epithelium, reproductive tissues and the sperm flagella [24]. Mutations and subsequent aberrant functioning or loss of CEP164 may contribute to lung phenotypes such as pulmonary bronchiectasis, as seen in some patients with *CEP164* mutations [1]. This may also contribute to aberrant spermatid function, which may cause infertility problems as seen typically in BBS models [9].

CEP164 is also expressed in other developing tissues that contain a primary cilium (human 8 PCW, mouse P15.5–30.5), including cardiomyocytes, cholangiocytes, tracheal and costal

cartilage, smooth muscle cells of the trachea, bronchi, bronchioles and seminiferous tubules, spermatogonia, spermatocytes, bone primordia and the epithelium lining and smooth muscles of the GI tract [69–76]. Notably, CEP164 is not present in hepatocytes, which do not contain primary cilium [77].

CEP164 is expressed in tissues involved in the CEP164 NPHP-RC phenotype, but is also expressed in tissues not associated with this phenotype. It could be that CEP164 has cell specific functions, or that not all tissues require primary cilia for development or that cells have other pathways that can compensate for the abnormal functioning or loss of CEP164.

Interestingly, the patterns of human CEP164 expression described here correlate with those of other JBTS genes *AHI1* and *CEP290* [78]. This could indicate a universal mechanism that underlies NPHP-RC.

In summary, CEP164 demonstrates widespread yet defined expression throughout human and murine development, which is predominantly maintained into adult life. Human and murine data largely correlate and CEP164 function is likely to be conserved between the two species. CEP164 is expressed in tissues affected in CEP164-NPHP-RC patients, however clinical heterogeneity, commonly seen in ciliopathies, requires further investigation.

## Supporting information

**S1 Table. Working dilutions of primary antibodies used for immunohistochemistry of human tissues.**
(DOCX)

**S2 Table. Comparison of human and murine kidney developmental timeline.**
(DOCX)

**S3 Table. Comparison of human and murine retina developmental timeline.**
(DOCX)

**S4 Table. Comparison of human and murine cerebellar development timeline.**
(DOCX)

**S1 Fig. Conservation and protein domains of human CEP164.** Predicted human CEP164 protein domains of the common 1460bp isoform; tryptophan-tryptophan (WW) domain conserved with two Tryptophan (W) residues, Lysine-rich repeat (LR) and predicted coiled-coil (CC) domains. Values marked are amino acid number (A). Sequence alignment of human CEP164 and its orthologs in *M.musculus*, *D.melanogaster*, *C. reinhardtii* and *D rerio* (B). Unrooted phylogenetic tree of CEP164 orthologs (C).
(TIF)

**S2 Fig. Diagram of the 129/OlaHsd- *Cep164^tm1a* allele.** Upon pre-mRNA splicing of *Cep164^tm1a(EUCOMM)Wtsi*, exon 3 splices into the splice acceptor (SA) of the *LacZ* cassette, causing a frameshift and subsequent formation of a premature termination codon, this forms the tm1b allele. The *LacZ* has an internal ribosomal entry site (IRES), and thus the *LacZ* fusion gene acts as a reporter gene for *Cep164*. Figure adapted from MRC Harwell International Mouse Phenotyping Consortium.
Internal ribosomal entry site (IRES), splice acceptor (SA), polyadenylation site (pA).
(TIF)

**S3 Fig. Digital magnification of human kidney CEP164 expression.** Human kidney 14 PCW (A), 18 PCW (B). CEP164 is expressed strongly at the apical membrane of the ureteric bud (A. I, B.I) which is maintained in the collecting duct, with CEP164 expression also at the

basolateral membrane (A.II,B.II). CEP164 is also expressed strongly at the renal tubule apical membrane (A.III, B.III).

Cap mesenchyme (CM), collecting duct (CD), loop of Henle (LH), renal tubule (RT), ureteric bud (UB).

(TIF)

**S4 Fig. Human CEP164 expression controls.** No primary antibody controls in the human 8 PCW kidney (A) and human 18 PCW kidney (B). Renal vesicles, comma-shaped vesicles, s-shaped body, ureteric bud and cap mesenchyme demonstrate no endogenous peroxidase staining (A.I.II.III, B.I.II.III). Renal tubule and the renal corpuscle (A.IV.V.VII, B.IV.V.VI), loop of Henle and collecting ducts (B.VII) also demonstrate no endogenous peroxidase staining. No primary antibody controls 8 PCW embryo (C). Lung, including the bronchioles and alveoli (C.I.II.III) and cardiomyocytes (C.I.IV,V) show no endogenous peroxidase staining. There is very weak endogenous staining in the Muscularis Externa (ME) of the gastrointestinal tract (C. VI) and the seminiferous cord of the gonads (C.VII.VIII). Dorsal root ganglia (C.IX) and bone primordia (C.VX) demonstrate no endogenous staining. However, the hepatocytes of the liver have endogenous peroxidase staining (C.IX), which means it cannot be determined if CEP164 is expressed in hepatocytes. The developing brain no primary controls, 8PCW (D, E) and 16 PCW (F). The developing brain (D.I, E.I) including the telencephalon (D.VI), metencephalon (D.V), midline (E.II) and myencephalon (D.VIII) do not demonstrate endogenous peroxidase staining. The neuroepithelium surrounding the brain ventricles (D.VIII, E.III.IV, F.I.III.IV) alongside the choroid plexus ependymal cells (D.II.III.IV, F.I.V) and choroid plexus pia matter (D.III.IV, F.I.V) do show some weak endogenous peroxidase staining. The molecular cell layer, purkinje and ganglion cell layers of cerebellum (F.I.II) and cerebellar white matter (F. VII) do not show endogenous staining. The ventricular surface (F.III.IV), medulla oblongata (F.VI) and choroid plexus at 16 PCW (F.V) show weak CEP164 staining. PAX6 positive control antibody (E.V, F.IX). The developing retina no primary controls, 11 PCW (G), and 19 PCW (H), both demonstrate no endogenous staining.

Alveoli primordia (Av Pri), bone primordia (BP), bronchiole (Br), cap mesenchyme (CM), cardiomyocyte (CM), cerebellum (Ce), choroid plexus (CP), collecting duct (CD), cortical plate (CorP), diencephalon (Di), dorsal root ganglia (DRG), ependymal cells (Ep), ganglion cell layer (GCL), gastrointestinal tract (GI), germline epithelium (GE), gonad (Go), intermediate Zone (IL), heart (Ht), hepatocytes (Hep), inner plexiform layer (IPL), inner nuclear layer (INL), fourth ventricle (FV), kidney (Ki), lateral ventricle (LV), lamina propria (LM), liver (Li), loop of Henle (LH), lung (Lu), marginal Layer (MaL), medulla oblongata (MO), mesencephalon (Mes), metencephalon (Met), midline (Mi), molecular Cell Layer (MCL), muscularis externa (ME), muscularis mucosae (MM), myencephalon (Mye), nasal epithelium (Na) nerve fibre layer (NFL), outer nuclear layer (ONL), outer neuroblastic cell layer (ONBL), outer plexiform layer (OPL), pia (P), photoreceptor cells (PS), post conception weeks (PCW), purkinje cell layer (PCL), renal corpuscle (RC), renal tubules (RT), renal vesicle (RV), retinal pigment epithelium (RPE), seminiferous cord (SC), s-shaped body (SSB), submucosae (SB), sub-ventricular layer (SVL), squamous epithelium mucosae (Mc-sq), telencephalon (Tel), ureteric bud (UB), ventricular layer (VL), ventricular surface (VS), white matter (WM).

(TIF)

**S5 Fig. Littermate wildtype controls for murine *Cep164* expression.** Murine retina wildtype controls (A). Murine kidney wildtype controls (B-C), murine secondary tissues wildtype controls (D-F), and murine cerebellar tissues (G-I). Retina at P1.5 (A.I), P15.5 (A.II), and P29.5 (A.III) demonstrate no endogenous beta galactosidase staining. Renal vesicles (B.I.II.III), ureteric bud (B.III), renal tubules (B.IV) and renal corpuscle (B.V.VI) at P1.5 do not show any

endogenous staining. Renal tissue at P30.5 have low endogenous beta galactosidase expression in the renal tubules (C.I.III.IV), but not in the renal corpuscle (C.II). Developing brain (D.I), costal cartilage (D.II) and intercostal muscle (D.III) at P0.5 do not show endogenous beta galactosidase staining. The P0.5 lung bronchioles (E.I.II.III.IV) demonstrate weak endogenous beta galactosidase staining, but not the alveoli (E.V). The P30.5 murine lung trachea, bronchioles and alveoli (F.I.II.III.IV.V.VI) do not show endogenous beta galactosidase staining. Murine cardiomyocytes (E.VI.VII, F.VI.VII), developing testes including the spermatogonia, spermatocytes, spermatids and smooth muscle cells (E.VIII.IX.X, F.VIII.IX.X) do not have endogenous staining. There is also no endogenous staining in the hepatocytes of the liver, hepatic portal vein (E.XI.XII.XIII, F.XI.XII.XIII). Murine brain cerebral hemisphere (G.I.V, H.I.V, I.I.), the midbrain (G.VI,H.VII,I.V), the striatum (H.V,) the thalamus (H.VI,I.VI) the pons and the medulla oblongata (G.I.VII, H.I.VII, I.I.VII) all demonstrate no endogenous staining. The choroid plexus shows some weak endogenous beta galactosidase expression (G.I.II.IV) at P0.5. In the developing the cerebellum, the ganglion cell layer, molecular cell layer, purkinje cell layer and white matter of the cerebellum (G.II.III, H.II.III.IV, I.II.III.IV) demonstrate no endogenous beta galactosidase staining.

Alveoli (Av), bronchiole (Br), cardiomyocyte (CM), cartilage (Ca), cerebellum (Ce), cerebral hemisphere (Ch), cerebrum (Cr), choroid plexus (CP), connective tissue (CT), costal cartilage (CC), epithelial (Ep), ganglion cell layer (GCL), heart (Ht), hepatocytes (Hep), inner nuclear layer (INL), inner plexiform layer (IPL), intercostal muscle (IC), liver (Li), lung (Lu), marginal layer (MaL), medulla oblongata (MO), midbrain (MB), molecular cell layer (MCL), outer neuroblastic layer (ONBL), outer nuclear layer (ONL), outer plexiform layer (OPL), photoreceptor segment layer (PS), pons (PO), postnatal day (P), purkinje cell layer (PCL), renal corpuscle (RC), renal tubule (RT), renal vesicle (RV), respiratory bronchiole (R), retinal pigment epithelium (RPE), seminiferous tubule (ST), smooth muscle (SM), spermatids (Sp), spermatocytes (SC), spermatogonia (Spg), striatum (St), terminal bronchi (TB), thalamus (Th), trachea (Tr), ureteric bud (UB), white matter (WM).

(TIF)

**S6 Fig. CEP164 wholemount expression throughout murine embryonic development and corresponding WT littermate controls.** At E9.5 (A), *Cep164* widespread expression is seen, including the branchial arches, developing forebrain, midbrain and hindbrain (A.I). There is also *Cep164* expression seen in the developing neural tube, including the neuroepithelium surrounding the anterior neuropore (A.II). There is *Cep164* expression seen in the developing heart including the central ventricle, bulbous cordis and outflow tract (A.III), as well as the developing limb buds (A.IV). This *Cep164* expression pattern is maintained at E10.5, with *Cep164* expression also seen in the optic cup and olfactory pit (B). At E12.5 *Cep164* expression is widespread but is defined in the spinal cord, vertebrae, cerebral cortex, midbrain, hindbrain, pons and medulla oblongata, otocyst and heart (C.I.II.III). *Cep164* expression is seen in the retina (C.IV) and at the tips of the developing digits, where the apical ectodermal ridge is present (C.V.VI.VII.VIII). WT littermates do not show endogenous beta galactosidase expression at E9.5 (D.I), E10.5 (D.II), and E12.5 (D.III).

Anterior nucleopore (ANP), apical ectodermal ridge (AER), branchial arch (BA), bulbous cordis (BC), cerebral cortex (CC), common atria (CA), common ventricle (CV), forebrain (FB), forelimb bud (FL), heart (Ht), hindbrain (HB), hindlimb bud (HL), medulla oblongata (MO), midbrain (MB), neural tube (NT), olfactory pit (OF), optic cup (OC), organ primordia (Pri), otocyst (OT), outflow tract (OFT), pons (PO), somites (SM), spinal cord (SC), vertebral column (VC).

(TIF)

**S7 Fig. Digital magnification of murine *Cep164* testes expression.** P15.5 murine testes (A) P30.5 murine testes (B). *Cep164* is expressed in the connecting tubules and leydig cells (A.I.II). In the seminiferous tubule, *Cep164* is expressed in the smooth muscle cells, spermatogonia and spermatocytes (A.B). *Cep164* is expressed most strongly in the spermatid, tails (A.III). Connecting tubule (CT), Leydig cells (LD), seminiferous tubule (ST), smooth muscle cells (SM), spermatocytes (SC), spermatogonia (Spg), sperm (Sp).
(TIF)

**S8 Fig. Validation of CEP164 antibody using hURECs.** Representative image of human urine derived renal epithelial cells (hURECs) with rabbit anti- CEP164 staining (Human Protein Atlas, HPA37606) (A). CEP164 (green) can be seen at the base of the primary cilium, stained with mouse anti-ARL13B (Proteintech, 66739-1-1g) (red), correlating with CEP164's mature centriole localisation (A.I.II). Vectashield with DAPI (blue) stained cell nuclei. No primary antibody controls show no staining for CEP164 (B).
(TIF)

## Acknowledgments

L.A.D is funded by the Medical Research Council Discovery Medicine North Doctoral Training Partnership, and The Northern Counties Kidney Research Fund, S.A.R is supported by a Kidney Research UK Post-doctoral fellowship (PDF_003_20151124). JAS is funded by Kidney Research UK and The Northern Counties Kidney Research Fund.

## Author Contributions

**Conceptualization:** J. A. Sayer.

**Data curation:** L. A. Devlin.

**Formal analysis:** J. A. Sayer.

**Funding acquisition:** C. G. Miles, J. A. Sayer.

**Investigation:** L. A. Devlin, S. A. Ramsbottom, L. M. Overman, E. Molinari, L. Powell, J. A. Sayer.

**Methodology:** L. M. Overman, S. N. Lisgo, G. Clowry, E. Molinari, J. A. Sayer.

**Project administration:** L. M. Overman, C. G. Miles, J. A. Sayer.

**Resources:** J. A. Sayer.

**Supervision:** S. A. Ramsbottom, S. N. Lisgo, G. Clowry, E. Molinari, C. G. Miles, J. A. Sayer.

**Writing – original draft:** L. A. Devlin, J. A. Sayer.

**Writing – review & editing:** L. A. Devlin, S. A. Ramsbottom, L. Powell, J. A. Sayer.

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
