## [Decision Letter · Decision Letter 0]

17 Sep 2019

PONE-D-19-22989

Embryonic and foetal expression patterns of the ciliopathy gene CEP164

PLOS ONE

Dear Prof Sayer,

Thank you for submitting your manuscript to PLOS ONE. After careful consideration, we feel that it has merit but does not fully meet PLOS ONE’s publication criteria as it currently stands. Therefore, we invite you to submit a revised version of the manuscript that addresses the points raised during the review process.

Please validate your findings by demonstrating that the primary antibody for CEP164 is specific (see, example the suggestions in *Nature Methods*
**volume 13**, pages 823–827 (2016) or by employing a second primary antibody that was produced independently. Please improve your figures as recommended by the second reviewer.

We would appreciate receiving your revised manuscript by Nov 01 2019 11:59PM. To enhance the reproducibility of your results, we recommend that if applicable you deposit your laboratory protocols in protocols.io, where a protocol can be assigned its own identifier (DOI) such that it can be cited independently in the future. For instructions see: http://journals.plos.org/plosone/s/submission-guidelines#loc-laboratory-protocols

We look forward to receiving your revised manuscript.

Kind regards,

Alfred S Lewin, Ph.D.

Academic Editor

PLOS ONE

Journal Requirements:

Additional Editor Comments (if provided):

Reviewers' comments:

Reviewer's Responses to Questions

**Comments to the Author**

1. Is the manuscript technically sound, and do the data support the conclusions?

Reviewer #1: Yes

Reviewer #2: Partly

2. Has the statistical analysis been performed appropriately and rigorously? 

Reviewer #1: N/A

Reviewer #2: N/A

3. Have the authors made all data underlying the findings in their manuscript fully available?

Reviewer #1: Yes

Reviewer #2: Yes

4. Is the manuscript presented in an intelligible fashion and written in standard English?

Reviewer #1: Yes

Reviewer #2: Yes

5. Review Comments to the Author

Reviewer #1: 1. Summary of the research

In this manuscript, Devlin et al. examine the expression patterns of the centrosomal protein CEP164 during development in human and mouse tissues. They find that the expression is widespread in different organs and tissue regions, and that the expression in mouse matches that in human to a large extent, with some differences.

CEP164 has been shown to play a role in Nephronophthisis-related ciliopathies (NPHP-RC), a group of genetic disorders with has multiple clinical manifestations. The expression of CEP164 in multiple organs throughout development supports its role in the multiple phenotypes seen in NPHP-RC.

The immunohistochemistry stainings are convincing, in a range of tissues and developmental stages. The data is very informative, but qualitative.

It would help to compare respective levels of expression if the authors quantified the intensity of staining in the different regions and sub-regions investigated, especially when looking at decreasing expression with time. This might be informative when looking at specific defects associated with particular areas within tissues.

Overall this is a really nice piece of work examining CEP164 expression in human and mouse embryonic and foetal tissues. I recommend publication

2. Examples and evidence

Major issues

Please could you quantify expression or adjust text to say that ‘expression seems to be lost with maturity’, for example.

Minor issues

Please amend the following:

• Line 74: ‘in some patients’, instead of ‘is some patients’

• Line 92: what is OFD IX? Maybe replace by Orofaciodigital syndrome 9?

• Fig S1, D: not sure how this panel shows absence of CEP164 in nematoda taxa? Confusing diagram. Is there a way of labelling directly on the graph maybe?

• Line 275: should it be C.III instead of C.II for S-shape body?

• Fig1, panel E: inset for E.II is rotated 90 degree to the right, could you put it in the correct orientation please?

• Inset for E.III does not correspond to the black box on panel E.I, could you adjust the black box please?

• Black box for panel E.IV is missing, could you add it please?

• Line 283-284: Fig1: there doesn't seem to be a panel F. Please amend text, or amend Fig 1 to add panel F.

• Line 327: text mentions Inner Segment and Outer Segment of photoreceptor layer. Fig 2A.III does not mention these, please amend.

• Line 525: there is no panel D in Fig 4. Please amend text or add panel D. This may be a confusion in panel lettering, as 15.5 is panel B and 30.5 is panel C. So the correct notation could be

(Figure 4B.XV.XVI.XVII, C.XIII.XIV), instead of (Figure 4C.XV.XVI.XVII, D.XIII.XIV).

3. Other points (optional)

I will be available to look at a revised version.

Reviewer #2: Devlin et al report embryonic and fetal expression patterns of the ciliary gene CEP164 in human and mouse specimens. CEP164 is an important distal appendage protein that localizes to the distal area of centrioles/basal bodies and its function is essential for ciliogenesis. Mutations in this gene cause NPHP-RC.

The authors performed immunohistochemistry for human embryonic sections and X-gal staining for mouse embryonic sections to conduct a thorough analysis of CEP164 expression. Similar expression patterns were observed between human and mouse tissues. CEP164 is widely expressed yet show some defined expression in multiple organs.

The study is thoroughly done and detailed expression results are provided. However, errors and mislabeling in figures make is extremely hard to read through. Although it provides new expression sites of CEP164 in embryos, it is descriptive in nature and would be suitable for a more specialized journal. Some specific points are listed below.

1) Human embryonic expression studies rely on only one antibody (Sigma). I know this has been used in several studies mostly for cultured cells, it does not necessarily mean that it is specific on tissues. I recommend to use another antibody for validation. Related to this, it would be helpful to show CEP164 localization at the base of cilia for antibody specificity as high mag images of trachea or airway for example.

2) Fig1: several rectangles for high magnification are missing. For example, A.VII, B.VII, C.VII, D.II, and E.IV. In some cases, rectangles are not accurately placed for high mag images. For example, A.V and AVI. Please also check Fig4 A.VI.

3) Fig1: it is described that CEP164 expression is seen in the cells of the uteric bud at both the apical and basal membranes as well as at the basolateral membrane of the collecting duct tubule. It is novel and interesting. However, these are very difficult to see. High mag images would be helpful. There are other places where high mag images might be beneficial such as CEP164 expression in sperm tail in Fig4.

4) In some cases where CEP164 expression is weak, for example, in NFL (Fig2A), it is hard to distinguish weak expression vs background. Perhaps, side by side comparison with negative controls may help clarify this.

Here are some errors or suggestions in the text. There are many more errors especially figure numbers so please carefully check:

P4 line 89: “basal body” to “centriole”

line 95: “Proteomics analysis” to “Protein domain analysis”

line 96: “serine/glutamine” to “serine-glutamine/threonine-glutamine”

P7 line 160: “1:1” to “1:3”

P12 line 274: remove “is”

line 275: “C.II” to “C.III”

line 283: “F.IV” to “E.IV”

line 284: “D.V” to “D.V, D.Vl” (CEP164 seems to be expressed in D.Vl)

line 284: “F.IV” to “E.II”

P14 lines 309 and 312: “P29.5” to “P30.5”

6. PLOS authors have the option to publish the peer review history of their article (what does this mean?). If published, this will include your full peer review and any attached files.

Reviewer #1: No

Reviewer #2: No

---

## [Author Response · Author response to Decision Letter 0]

20 Nov 2019

PONE-D-19-22989

Embryonic and foetal expression patterns of the ciliopathy gene CEP164

PLOS ONE

Dear Dr Lewin, 

Thank you for your comments regarding our manuscript “Embryonic and foetal expression patterns of the ciliopathy gene CEP164”. We have made amendments to the manuscript, to address the points generated in the review. Please see the revised copy of the manuscript with tracked changes, and the revised manuscript. 

1) In response to “Please validate your findings by demonstrating that the primary antibody for CEP164 is specific” and “Human embryonic expression studies rely on only one antibody (Sigma). I know this has been used in several studies mostly for cultured cells, it does not necessarily mean that it is specific on tissues. I recommend to use another antibody for validation. Related to this, it would be helpful to show CEP164 localization at the base of cilia for antibody specificity as high mag images of trachea or airway for example”. 

We have added a supplementary figure (S8), demonstrating that the rabbit anti-CEP164 is defined to the base of the primary cilium of human urine derived renal epithelial cells (hURECs), which are phenotypically relevant to this study and the CEP164-NPHP patients. Additionally we would like to emphasise that the alternative non-immunohistochemical LacZ reporter assay used in the murine Cep164 expression studies further validates the human embryonic and foetal CEP164 antibody staining, due to correlating expression patterns. We therefore feel that CEP164 expression in the human embryo and foetal tissues is validated. 

2) Reviewer 1s comment “Please could you quantify expression or adjust text to say that ‘expression seems to be lost with maturity” was thoroughly considered. 

It was decided that quantification of CEP164 staining to compare against different time-points and tissue sub-regions could not be completed accurately enough due to the semi-quantitative nature of DAB immunohistochemistry and LacZ assays. Hence, the manuscript was altered so that quantification statements were not made. 

3) In response to reviewer 2s comments “In some cases where CEP164 expression is weak, for example, in NFL (Fig2A), it is hard to distinguish weak expression vs background. Perhaps, side by side comparison with negative controls may help clarify this”. 

Representative controls have been added to Fig 1, Fig 2 and Fig 3, to offer side-by-side comparisons. Supplementary figures S4 and S5, provide all of the controls for further analysis. Additionally, human retinal no-primary controls have been completed, to examine for endogenous peroxidase activity, and have been added to Fig 2 and S4. 

4) As highlighted by reviewer 1 “Fig S1, D: not sure how this panel shows absence of CEP164 in nematoda taxa? Confusing diagram. Is there a way of labelling directly on the graph maybe?” 

We acknowledge that the S1 Fig D diagram was confusing. It has been decided that it does not add sufficient information to the manuscript, and has therefore been removed. 

5) All grammatical, spelling and formatting errors highlighted by the reviewers, and found within the manuscript have been resolved to the best of our knowledge. All of the Figure insets have been adjusted accordingly so that they are in the correct orientation, figures are correctly labelled and the figure panels match up with the text. 

6) Reviewer 1 stated that “Black box for panel E.IV is missing, could you add it please?”, in regards to Figure 1. Additionally, reviewer 2 stated that “Fig1: several rectangles for high magnification are missing. For example…….B.VII, C.VII….and E.IV.” 

There has been a misunderstanding here. The rectangles are not required in these locations due the images not being a high magnification, they are from separate medullary region of the developing kidney. This has now been clearly labelled in the figure, and figure legends. 

7) Reviewer 1 stated that “Line 327: text mentions Inner Segment and Outer Segment of photoreceptor layer. Fig 2A.III does not mention these, please amend.” 

Fig 2A.III has now been labelled appropriately. 

8) To address the following point from reviewer 2 “Fig1: it is described that CEP164 expression is seen in the cells of the uteric bud at both the apical and basal membranes as well as at the basolateral membrane of the collecting duct tubule. It is novel and interesting. However, these are very difficult to see. High mag images would be helpful. There are other places where high mag images might be beneficial such as CEP164 expression in sperm tail in Fig4.” 

Supplementary figures S3 and S7 have been generated which show digital magnification images of CEP164 expression at the nephron tubule membranes and seminiferous tubules containing sperm, respectively. 

9) Reviewer 3 stated that “P7 line 160: “1:1” to “1:3””. 

I feel there has been misunderstanding here. The methods are describing the mating of a heterozygous CEP164tm1a mouse with a wild type mouse, therefore the ratio of HET:WT will indeed by 1:1, as we are trying to maintain the HET line and not generate HOM mice. Please see the punnet square below. 

Additionally, the discussion has been modified to be more clear and concise; amendments can be seen in the tracked changes file. 

The manuscript, and associated files, have been edited to meet PLOS ONE’s style requirements. All data is provided in the manuscript, and the data availability statement has been updated accordingly. 

Thank you for the helpful reviews.

---

## [Decision Letter · Decision Letter 1]

17 Dec 2019

PONE-D-19-22989R1

Embryonic and foetal expression patterns of the ciliopathy gene CEP164

PLOS ONE

Dear Prof Sayer,

Thank you for submitting your manuscript to PLOS ONE. After careful consideration, we feel that it has merit but does not fully meet PLOS ONE’s publication criteria as it currently stands. Therefore, we invite you to submit a revised version of the manuscript that addresses the points raised during the review process.

Reviewer 2 recommends some minor edits to the text and to Figs. 1 and 3. Please address these suggestions or make the changes requested.

We would appreciate receiving your revised manuscript by Jan 31 2020 11:59PM. To enhance the reproducibility of your results, we recommend that if applicable you deposit your laboratory protocols in protocols.io, where a protocol can be assigned its own identifier (DOI) such that it can be cited independently in the future. For instructions see: http://journals.plos.org/plosone/s/submission-guidelines#loc-laboratory-protocols

We look forward to receiving your revised manuscript.

Kind regards,

Alfred S Lewin, Ph.D.

Academic Editor

PLOS ONE

Reviewers' comments:

Reviewer's Responses to Questions

**Comments to the Author**

1. If the authors have adequately addressed your comments raised in a previous round of review and you feel that this manuscript is now acceptable for publication, you may indicate that here to bypass the “Comments to the Author” section, enter your conflict of interest statement in the “Confidential to Editor” section, and submit your "Accept" recommendation.

Reviewer #1: All comments have been addressed

Reviewer #2: All comments have been addressed

2. Is the manuscript technically sound, and do the data support the conclusions?

Reviewer #1: Yes

Reviewer #2: Yes

3. Has the statistical analysis been performed appropriately and rigorously? 

Reviewer #1: N/A

Reviewer #2: N/A

4. Have the authors made all data underlying the findings in their manuscript fully available?

Reviewer #1: Yes

Reviewer #2: Yes

5. Is the manuscript presented in an intelligible fashion and written in standard English?

Reviewer #1: Yes

Reviewer #2: Yes

6. Review Comments to the Author

Reviewer #1: The authors have satisfcatorily addressed all the reviewer's comments, thank you very much to them. The paper is clearer and the addition of controls provides confidence in the stainings. I recommened publication.

Reviewer #2: The manuscript has been improved significantly. I have a few more edits to suggest:

line 100: remove "."

line 126: musculus

line 171: addition

line 231: glass

line 250: please add a reference here

line 265: was

Fig 1B V and VII are not properly cropped so please check

Letters in some figures are very hard to see for example Fig 3A VI and B VI.

The following paper can be cited:

Airik et al., Roscovitine blocks collecting duct cyst growth in Cep164-deficient kidneys, Kidney Int, 2019

7. PLOS authors have the option to publish the peer review history of their article (what does this mean?). If published, this will include your full peer review and any attached files.

Reviewer #1: No

Reviewer #2: No

---

## [Author Response · Author response to Decision Letter 1]

3 Jan 2020

The minor comments have all been addressed in a revised version.

---

## [Editor Report · Decision Letter 2]

6 Jan 2020

Embryonic and foetal expression patterns of the ciliopathy gene CEP164

PONE-D-19-22989R2

Dear Dr. Sayer,

We are pleased to inform you that your manuscript has been judged scientifically suitable for publication and will be formally accepted for publication once it complies with all outstanding technical requirements.

With kind regards,

Alfred S Lewin, Ph.D.

Section Editor

PLOS ONE
---

## [Editor Report · Acceptance letter]

16 Jan 2020

PONE-D-19-22989R2 

Embryonic and foetal expression patterns of the ciliopathy gene *CEP164*

Dear Dr. Sayer:

I am pleased to inform you that your manuscript has been deemed suitable for publication in PLOS ONE. Congratulations! Your manuscript is now with our production department. 

With kind regards,

on behalf of

Dr. Alfred S Lewin 

Section Editor

PLOS ONE